# Association Between Childhood Obesity and the Risk of Food Addiction: A Matched Case-Control Study

**DOI:** 10.3390/nu17162654

**Published:** 2025-08-15

**Authors:** Néstor Benítez Brito, Berta Pinto Robayna, Juan Ignacio Capafons Sosa, Miguel Angel García Bello, Eva María Herrera Rodríguez, Jesús Enrique de las Heras Roger, Mónica Ruiz Pons, Irina María Delgado Brito, Carlos Díaz Romero, Yolanda Ramallo Fariña

**Affiliations:** 1Department of Chemical Engineering and Pharmaceutical Technology, Nutrition and Bromatology Area, Pharmacy Faculty, Universidad de La Laguna, 38206 San Cristóbal de La Laguna, Tenerife, Spain; extbpintoro@ull.edu.es (B.P.R.); jherasro@ull.edu.es (J.E.d.l.H.R.); irinadb@hotmail.es (I.M.D.B.); cdiaz@ull.edu.es (C.D.R.); 2Nutrition, Health and Food Research Group (NAYS), Universidad de La Laguna, 38206 San Cristóbal de La Laguna, Tenerife, Spain; 3Canary Foundation Institute of Health Research of the Canary Islands (FIISC), Evaluation Service of the Canary Islands Health Service (SESCS), 38109 Santa Cruz de Tenerife, Tenerife, Spain; ji.capafons.sescs@gmail.com (J.I.C.S.); miguel.garciabello@sescs.es (M.A.G.B.); yolanda.ramallofarina@sescs.es (Y.R.F.); 4Department of Pediatrics, Hospital University Nuestra Señora de Candelaria, 38010 Santa Cruz de Tenerife, Tenerife, Spain; hr_eva@hotmail.com (E.M.H.R.); mruizpon@ull.edu.es (M.R.P.); 5Network for Research on Chronicity, Primary Care, and Health Promotion (RICAPPS), 38109 Santa Cruz de Tenerife, Tenerife, Spain

**Keywords:** feeding and eating disorders, food addiction, mental disorders, obesity, pediatric obesity

## Abstract

**Background:** Food addiction is a new clinical entity that is beginning to be linked to obesity and eating disorders. The present study aims to investigate the association between the risk of food addiction in children and the presence of obesity. It also explores the relationship between food addiction, the development of eating disorders, and body image dissatisfaction. **Material and methods:** A matched case-control study was conducted in a Spanish pediatric population (cases have obesity, and controls have normal weight). The main outcome measures were evaluation of food addiction (S-YFAS-C), child feeding attitudes (ChEAT), and evaluation of body image (CDRS). Additionally, sociodemographic and anthropometric data were gathered. **Results:** A total of 62 children were evaluated (31 cases with age 11 ± 0.7 years and BMI Z-score 2.89 ± 1.33; 31 controls with age 10.7 ± 0.8 years and BMI Z-score −0.05 ± 0.52). For all items on the S-YFAS-C scale, significant differences were observed between the two groups (∧ = 0.252, *p* = 0.002). Food addiction was diagnosed in 32.3% of cases (2.06 ± 1.7 symptoms) and 22.6% of controls (1.61 ± 1.6 symptoms), although no statistically significant differences were observed between groups. A statistically significant correlation exists between all the scores of the scales studied in the children. **Conclusions:** Children with obesity have a higher number of food addiction symptoms compared to those with normal weight. In general, as food addiction scores increase, higher scores are observed for the risk of developing eating disorders and body image dissatisfaction.

## 1. Introduction

In the last decade, the theory of food addiction (FA) has taken hold in society, and numerous studies have attempted to respond to this new clinical entity [1]. It is a pathology characterized by an eating behavior based on the high consumption of certain foods in an apparently uncontrollable way [2]. However, this concept is not yet officially recognized in the Diagnostic and Statistical Manual of Mental Disorders (DSM) [3] or in the International Classification of Diseases (ICD-11) [4]. Although there are indications that certain foods may be associated with binge eating, further research is still needed to determine whether food addiction can be classified as a behavioral disorder, similar to pathological gambling, or as a form of dependence analogous to substance abuse [5].

In itself, this new model of FA displays behavioral patterns similar to substance use disorders, challenging traditional views on eating behavior and dietary self-regulation. FA is characterized by compulsive eating and an inability to regulate food intake despite negative consequences [6].

Currently, the Yale Food Addiction Scale tool allows the assessment of FA through a questionnaire that operationalizes the concept of FA by adapting the diagnostic criteria of substance-related disorders outlined in the DSM-V [3] and their application to eating behavior [7]. This tool has been adapted for various populations and translated into different languages in order to investigate this exploratory notion [8,9,10,11]. By adapting questions according to the target population, a wide range of criteria related to the consumption of fatty, high-calorie, and sugary foods is used. Some of these questions focus on the diagnosis of addiction, but others are more oriented toward the identification of symptoms.

Recent research has found that patients with FA traits may be more susceptible to developing Eating Disorders (EDs), such as bulimia nervosa, binge eating disorder, and anorexia nervosa [12]. This connection underlines a complex interplay between psychological, biological, and environmental factors that may predispose individuals to these disorders [13]. Furthermore, the prevalence of FA appears to correlate with increased rates of overweight and obesity, calling for further research into how addictive behaviors around food consumption may exacerbate or contribute to the development of eating disorders and obesity [14].

The onset of ED commonly occurs in preadolescence and adolescence [15]; the marked physiological and morphological changes that occur at puberty are sometimes accompanied by maladaptive developmental trajectories among young people, which may explain both the high rates of subclinical forms of ED and their persistence over time [16]. The pediatric population is more vulnerable to these EDs than the adult population, and the relationship of FA with other already defined disorders begins to be explored.

Global studies on the prevalence of FA in children are increasing, although there is still a need to explore its causes and establish its risk factors [17]. In the Spanish population, few studies have examined this phenomenon in the pediatric population; however, in 2021, the only validated tool in Spanish was developed to measure FA in the pediatric population [18].

The present study aims to examine the association between the risk of FA in childhood and the presence of obesity. Additionally, the relationship between FA and the occurrence of ED and body image dissatisfaction is assessed as a function of the presence of obesity. Finally, the relationship between obesity and ED and body image dissatisfaction is also explored. We hypothesize that there is a relationship between the risk of FA and the presence of obesity. In addition, FA may be related to an increased risk of developing ED.

## 2. Methods

### 2.1. Design and Participants

A matched case-control study was conducted in a Spanish pediatric population.

The inclusion criteria for the cases were: (1) age between 9 and 12 years and (2) having a Body Mass Index (BMI) Z-score ≥ 2 SD (obesity). Patients were recruited consecutively when they first attended the nutrition clinic of the Pediatrics Department of the Hospital Universitario Nuestra Señora de Candelaria (HUNSC), a tertiary referral hospital on the island of Tenerife, Spain.

The inclusion criteria for the controls were: (1) age between 9 and 12 years and (2) having a BMI Z-score between −1.00 and +0.99 SD (normal weight). These subjects were recruited from a private school in the province of Santa Cruz de Tenerife, Spain.

In both groups, the consent of their legal guardians was required for participation. All invited children returned the signed consent form and agreed to participate. Minors with reading comprehension difficulties, intellectual or cognitive disabilities, or any previously diagnosed psychiatric pathology were excluded.

### 2.2. Selection of Cases and Controls

The initially recruited sample comprised 71 eligible children with obesity (cases) and 45 eligible normal-weight children (controls). Because the number of controls was lower than the number of cases, we applied 1:1 matching based on age (maximum allowed difference: 1 year) to create comparable groups. This procedure resulted in a final analytic sample of 31 matched pairs (31 cases; 31 controls). Although the reduced number of matched controls may limit generalizability, age matching helped minimize confounding by developmental stage.

### 2.3. Outcome Measures

Questionnaire for the evaluation of food addiction (FA) in the child population (S-YFAS-C) [18]: The questionnaire consists of 25 items and examines food behavior over the past 12 months based on seven criteria for diagnosing substance dependence according to the DSM-IV-TR. FA is assessed in two different ways based on scores: first, by counting symptoms, providing a score that reflects the number of dependence symptoms based on the seven criteria; and second, by diagnosing addiction, evaluating whether a FA diagnosis can be confirmed. This diagnosis is made when three or more symptoms and significant clinical impairment or distress are present. In this study, all items of the S-YFAS-C were administered exactly as presented in the validated Spanish version, without any modifications to the content and wording. However, in order to ensure comprehension, the standard instructions provided by the original authors were read aloud to all participants by the researchers. During this reading, it was clarified that the questions referred specifically to the consumption of ‘junk food,’ understood as including sweets (e.g., candy, ice cream, chocolate, and cake), carbohydrates (e.g., white bread, rolls, pasta, and rice), salty snacks (e.g., chips, pretzels, and crackers), fatty foods (e.g., french fries, hamburgers, pizza, bacon, and steak), and sugary drinks (e.g., soda pop, juice, milkshakes, smoothies, and energy drinks).

Children Eating Attitudes Test (ChEAT) [16,19]: This 26-item tool estimates the prevalence of the risk of developing an ED, helping identify issues related to a frequent and stable concern about food, abnormal eating patterns, and attitudes at these ages. The questionnaire score ranges from 0 to 78, with a threshold of 20 or more points indicating a risk of developing an ED. A reliability coefficient of 0.85 (Cronbach’s alpha) was obtained in our sample.

Contour Drawing Rating Scale (CDRS) [20]: This visual scale evaluates body image and consists of nine contours of male and female figures. The contours increase in size with the score, and the degree of satisfaction or discrepancy index is obtained by calculating the difference between the ideal and current body image. Negative scores indicate a desire to lose weight, while positive scores reflect a desire to gain weight. A difference of two or more points was interpreted as significant body image dissatisfaction.

Anthropometric evaluation. Data on weight, height, and BMI were obtained by means of an approved bioimpedance scale (Tanita BC-418 model) and the respective percentiles and standard deviation.

For both groups, demographic data and questionnaires were self-administered. Anthropometric measurement was performed for the cases by the medical-nutritional team at the hospital, while for the controls, it was performed by dietitian-nutritionists at the school, using the same equipment and procedure to ensure consistency in the measurements.

### 2.4. Data Analysis

Mean and standard deviation were used to summarize quantitative variables, while frequency and percentages were used for categorical variables. For the outcome measures, the assumptions of normality, skewness, and kurtosis were evaluated.

Correlational analyses were performed using Spearman’s rank due to the nature of the study’s outcome measures.

Independent sample *t*-test analyses were used to compare outcomes between the normal weight group and the obese group. Chi-square tests were employed to evaluate the independence of categorical variables, with Fisher’s exact test used in cases when necessary. A one-way multivariate analysis of variance (MANOVA) was conducted to evaluate group differences across the set of items from the S-YFAS-C scale. Prior to the analysis, key assumptions were evaluated. The assumption of homogeneity of covariance matrices could not be formally tested with Box’s M test, as the number of observations in each group was smaller than the number of dependent variables. Multivariate normality of the residuals was evaluated using the Shapiro–Wilk test and visually inspected via a Chi-square Q-Q plot. Given the constraints on assumption testing, Pillai’s Trace was selected as the primary multivariate test statistic due to its superior robustness to violations of both homogeneity and normality.

Reliability was assessed using Cronbach’s alpha values. For every statistical test, a significance level of *p* < 0.05 was determined. Jamovi v2.23.8.0 software and R 4.2.1 were utilized for the analyses.

### 2.5. Ethics Approval and Consent to Participate

The study has the approval of the Drug Research Ethics Committee of the Canary Islands University Hospital Complex (code CHUNSC_2020_55). Likewise, it meets the requirements of the Declaration of Helsinki, the Council of Europe Convention on human rights and biomedicine, the UNESCO Universal Declaration on human rights, the protection of personal data and bioethics, Law 14/2007 of 3 July on Biomedical Research, and the requirements of Spanish legislation. Informed consent was obtained by the researchers themselves, following international recommendations. In the case of the controls, each family was given a printed informed consent form along with all the information about the study in a sealed envelope. A period of two weeks was estimated for the correct reception of all the justifications. For the children to participate in the study, the parents had to send the signed informed consent before the start of the trial.

## 3. Results

Table 1 shows demographic and anthropometric characteristics of the sample, divided by group. Regarding gender distribution, 61.3% of the sample were males. Minimum and maximum values were calculated in the overall sample for age (9.5 and 12.3 years). Minimum and maximum values were also calculated, divided by group, for the rest of the anthropometric variables. In the case of the control group, the values were as follows: weight (26.6, 49.0 kg), height (1.31, 1.59 cm), BMI (14.6, 19.4 kg/m^2^), and z-score for BMI (−0.97, 0.90). Case group values were as follows: weight (51.4, 110 kg), height (1.42, 1.69 cm), BMI (25.5, 39.3 kg/m^2^), and z-score for BMI (2.0, 6.23). Table 1 shows no statistically significant differences between groups in either gender distribution (*p* = 0.118) or age (*p* = 0.088).

Table 2 displays the sample’s scores in each outcome measure for each group. In relation to the results obtained for applying S-YFAS-C for the evaluation of FA, no significant differences were found, neither in the quantification of symptoms (*p* = 0.291) nor in the number of positive diagnoses for FA (*p* = 0.437). Significant differences were only found in the second (*p* = 0.004) and fifth (*p* = 0.027) criteria, which relate to Chronic desire or repeated attempts to stop consumption, and substance use despite knowledge of adverse consequences.

When a MANOVA was applied to all items of the S-YFAS-C scale, Box’s M test could not be computed due to an insufficient number of cases in each group relative to the number of dependent variables. Additionally, the multivariate Shapiro–Wilk test was significant (*p* < 0.001), and the visual inspection of a Chi-square Q-Q plot suggested the residuals followed a light-tailed distribution, indicating a violation of multivariate normality. Given these concerns, we opted to use Pillai’s Trace, a statistic known to be more robust under violations of MANOVA assumptions. Statistical differences were found between the two groups (V = 0.748, *p* = 0.002), and differences remained significant even after excluding the two items that showed effects in the opposite direction of our expectations (V = 0.67, *p* = 0.009); see Figure 1.

No statistically significant differences between the normal-weight and obesity groups in the risk of developing an ED (*p*-value = 0.168) or in their scores (*p* = 0.222). Statistically significant differences were found in body image discrepancy (*p* < 0.001) and body image dissatisfaction (*p* < 0.001). The same analyses were conducted using parametric tests, and no relevant differences were found compared to the non-parametric results.

Table 3 shows the body image dissatisfaction and ED scores based on group and the presence of an FA diagnosis according to S-YFAS-C scores. For children with normal weight, significant differences were found in ED scores based on the FA diagnosis (*p* < 0.001). Comparisons in body image distortion showed no significant differences (*p* = 0.19).

Obese individuals showed significant differences in body image dissatisfaction scores (*p* = 0.023) according to the S-YFAS-C diagnosis, but the result was not significant for ED scores (*p* = 0.185).

Comparing normal-weight individuals to those with obesity, significant differences were found in body dissatisfaction scores in those with FA diagnosis (*p* = 0.015), as well as those without a diagnosis (*p* < 0.001). No significant differences were found in the comparison between normal-weight and obese individuals in the ChEAT-26 scores.

Additionally, the scores of the ChEAT-26 (ED) based on the S-YFAS-C diagnosis outcome, differentiating between normal-weight and obese groups, are presented in Figure 2. A tendency for higher values on the ChEAT26 with diagnosis of FA can be observed in both groups (cases and controls), although the differences did not become statistically significant.

Table 4 shows the correlation values between the outcome measures separated by group. Assessing the general sample, a statistically significant correlation was found between all scale scores. Specifically, in relation to the ChEAT-26 score, a significant positive correlation was found with the number of FA-related symptoms (r = 0.35, *p* = 0.005), and a significant negative correlation was found with the body image discrepancy score (r = −0.27, *p* = 0.031). On the other hand, the number of FA symptoms also showed a significant negative correlation with the body image discrepancy score (r = −0.36, *p* = 0.005).

When evaluating the correlations, including only subjects with obesity, only the correlation between the number of symptoms and the body image discrepancy score remains significant (r = −0.40, *p* = 0.027). A near-significant correlation was found between the number of FA symptoms and the ChEAT-26 score (r = 0.311, *p* = 0.09), as well as between the ChEAT-26 score and the body image discrepancy score (r = −0.32, *p* = 0.08).

Evaluating exclusively the subjects with obesity, BMI z score did not significantly correlate with the symptom count (r = −0.003, *p* = *0*.99), the ED score (r = 0.09, *p* = *0*.63), or body image discrepancy (r = −0.25, *p* = *0*.17) (Table 5).

## 4. Discussion

Until now, FA had not been evaluated in Spain in the pediatric population with obesity. This is the first study to analyze not only FA but also its relationship with ED, as well as to compare the results in children with normal weight *versus* children with obesity.

In general, children with obesity (cases) have higher scores in all tools applied in this study (S-YFAS-C, ChEAT-26, and CDRS) compared to children with normal weight (controls), as shown in Table 2.

According to the S-YFAS-C scale, there are significant differences between groups when analyzing the 25 items of the scale, as shown in Figure 1. In relation to the diagnosis of FA, which is established when three or more symptoms are present and the presence of anxiety or significant clinical deterioration is observed, 32.3% of patients with obesity were diagnosed with FA, while 22.6% of children with normal weight presented this diagnosis. However, no statistically significant differences were observed between the two groups.

Beyond the diagnosis, it should be noted that in younger samples, such as children, the symptom count score option is recommended to establish the risk of FA, as it seems to be the most sensitive measure for assessing subclinical eating behaviors [8]. In this sense, children with obesity also present more symptoms of FA than those with normal weight (2.06 symptoms *versus* 1.61 symptoms, respectively), although this is not statistically significant.

While neither the total symptom count nor diagnostic rates differed significantly between groups, the MANOVA analysis did reveal a statistically significant difference in the overall pattern of responses to the S-YFAS-C items. This suggests that children with obesity may exhibit a distinct behavioral profile related to food consumption, even if those differences are not fully reflected in the scale’s cumulative scores.

It is important to consider that the S-YFAS-C scoring algorithm, although psychometrically robust, can be restrictive—especially in younger populations—as it requires the endorsement of multiple symptoms plus clinical distress to generate a diagnosis. This may lead to a floor effect, where the limited number of endorsed symptoms makes it difficult to detect subtle yet relevant behavioral differences between groups.

The use of MANOVA was intended to address this limitation by analyzing the multivariate response structure across all items, rather than relying solely on aggregate scores or binary diagnostic outcomes. The emergence of statistically significant differences through this approach suggests that meaningful divergences in food-related behavior may exist between groups, even if they do not yet reach established diagnostic thresholds. This aligns with the view that FA-related behaviors may begin to emerge gradually during childhood, before fully consolidating into diagnosable patterns. Furthermore, when analyzing the seven diagnostic criteria individually, the most notable differences were found in criterion 2 (persistent desire or repeated attempts to reduce intake) and criterion 5 (continued consumption despite adverse consequences). These behaviors may reflect the habitual intake of ultra-processed foods, designed with industrial formulations that intensely stimulate the brain’s reward circuitry, which could contribute to the development of an addictive response, while promoting excess weight due to their high caloric content [21]. These criteria are the ones that usually give the most positive results in studies, as suggested in 2016 by Imperatori C. et al. [22].

Although no data are available in Spain to compare these results, Santos-Flores JM et al. conducted a similar analysis in Mexico, using an adaptation of the S-YFAS-c scale in 448 children [23]. Indeed, pediatric patients with obesity had higher scores on the FA scale, as in the present study. This is also evident from the systematic review by Yekaninejad et al., who, after analyzing 6996 children, found that those with obesity tended to have higher scores on the FA scale [14].

At the same time, the data on the diagnosis of FA (22.60%) in children with normal weight in the present study are similar to other studies. For example, in the initial validation of the S-YFAS-C scale, a prevalence of FA of 20% was obtained in healthy children, with a mean symptom count of 1.67 ± 1.45 [24]. Yekaninejad et al., in their systematic review with meta-analysis, in the sample of 6996 participants discussed above, obtained a similar symptom count in healthy children (1.54–95% CI: 1.23–1.85) [14].

With respect to the ED risk score, no significant differences were found between the groups. Although the limited sample size is one of the main explanations, this lack of significant differences may, in turn, be due to the fact that having an average weight does not exclude having an ED. In some of the most prevalent EDs, such as anorexia or bulimia, excessive weight is not a defining characteristic (rather the opposite). Therefore, taking this into account, it is to be expected that high ED symptom scores would also be found in the normal-weight group. Nevertheless, almost 40% of the cases have an elevated score for the risk of developing an ED, which is higher than the percentage observed in the controls.

However, when only normal-weight subjects are analyzed (Table 3), significant differences are observed, with higher ED scores in individuals diagnosed with FA. Therefore, it seems that when a child is diagnosed with FA, he or she also presents a fairly high risk for an ED. There seems to be a significant correlation in children, in general, since the higher the number of symptoms in the S-YFAS-C, the higher the scores on the risk of developing an ED (ChEAT-26), as shown in Table 4. This situation raises the suspicion that FA behaves as a possible ED, independently of excess weight, presupposing that excess weight may be a consequence (late or not) of FA. This was also observed in the pilot study of the validation of the S-YFAS-C (higher number of symptoms in FA, higher number of symptoms for ED risk) [24].

However, while these correlations are statistically significant, the strength of the associations is modest, indicating that the relationship between FA symptoms, ED risk, and body dissatisfaction may be only partial or influenced by additional variables (not included in this study). Moreover, part of the observed correlation could be inflated or partly artifactual, as it may result from analyzing a dataset composed of two groups (obesity vs. normal weight) with markedly different means, which can artificially strengthen group-level associations. Importantly, the study did not control for potential confounding variables such as impulsivity, emotional regulation, or family and peer influences, which could contribute to or moderate the associations observed.

The relationship between FA and ED is beginning to be increasingly studied, although the clinical relevance of this connection is still in its infancy, making it difficult to draw definitive conclusions [25,26]. However, it can be stated that FA seems to be associated with higher BMI, low self-esteem, and impulsive and emotional eating behaviors, as reflected by several authors [13,27,28,29]. In addition, the relationship between FA and certain EDs, such as binge eating disorder, bulimia nervosa, and anorexia nervosa, is evident [12]. Several studies highlight this relationship, and there is a large overlap in terms of diagnosis and symptom severity [26,30,31].

In relation to dissatisfaction with body image, there are significant differences between groups, as shown in Table 2, with children with obesity having higher rates of dissatisfaction with body image than those with normal weight (83.87% vs. 9.68%, respectively). Furthermore, analyzing image dissatisfaction only in the obese group, it is observed that children with FA have a higher image dissatisfaction than those without FA, which is also statistically significant.

Body image dissatisfaction is a common symptom in ED and obesity. Moreover, it is more prevalent in children and young people, as attested by Eun-Ha Jung et al. after studying 41,124 high school students [32]. Almost half of the students with body image dissatisfaction were trying to lose weight. The data presented in this paper, considered high by the authors, reflect that body image is central to many aspects of children’s functioning, such as emotions, thoughts, behaviors, and relationships. In general, it is observed that both boys and girls wish to have a thinner image than the one they show, as also reflected by the original authors of the S-YFAS-C scale [24].

Finally, the present work has limitations given the inherent nature of the study itself. It is a case-control trial with a limited sample, and no formal sample size calculation was performed; the number of participants was determined by the availability of eligible cases during the recruitment period and the accessibility of the control group, which may affect the statistical power of the findings. Therefore, the conclusions drawn are primarily, and a study with the same approach but with a more representative sample should be conducted in order to confirm these initial results. Furthermore, it should be noted that the findings reflect an association between FA and ED; demonstrating a direct causal relationship is more complex and will require further analysis. In addition, potential confounding variables—such as impulsivity, emotional regulation, or family and social context—were not assessed in this study, which may have influenced the observed associations between FA, ED risk, and body dissatisfaction. Moreover, the sample was geographically limited to Tenerife (Spain) and restricted to children aged 9 to 12 years, which may reduce the generalizability of the findings to other populations, regions, or age groups. Additionally, the interpretation of the MANOVA results should be considered in light of two statistical limitations. First, due to sample size constraints, the assumption of homogeneity of covariance matrices could not be formally tested using Box’s M test. Second, the assumption of multivariate normality was violated. To address these issues, we based our inference on Pillai’s Trace, which is considered the most robust multivariate statistic under such conditions. Nevertheless, the results should be interpreted with caution, and future studies with larger samples are needed to confirm these findings. However, despite these limitations, it can be emphasized that this is the first study carried out in Spain assessing FA and its relationship with ED, developed in both pediatric clinical individuals (obesity) and healthy pediatric individuals (controls).

To conclude, it should be noted that the original questionnaire incorporates specific instructions that make continuous reference to junk food, including sweets, salty snacks, high-fat foods, sugary drinks, and carbohydrates. In the questionnaire used in our research and on our sample, the written mention of this type of food was omitted, opting instead to explain it verbally. This modification may have attenuated the differences between the groups. Finally, it is relevant to highlight that items 11 and 12 of the S-YFAS-C showed variations in the opposite direction to what was expected, which could be attributed to the lack of explicit mention of junk food in the instructions, resulting in a different interpretation of these items.

## 5. Conclusions

Children with obesity have higher food addiction scores as measured by the S-YFAS-C scale compared to children with normal weight. Although differences between groups are evident on the S-YFAS-C scale, further studies are needed to fully investigate the relationship between food addiction and obesity.

Overall, with the sample analyzed, the pediatric population with a higher number of food addiction symptoms tends to have higher scores associated with the risk of developing eating disorders and to show higher levels of body image dissatisfaction.

## Figures and Tables

**Figure 1 nutrients-17-02654-f001:**
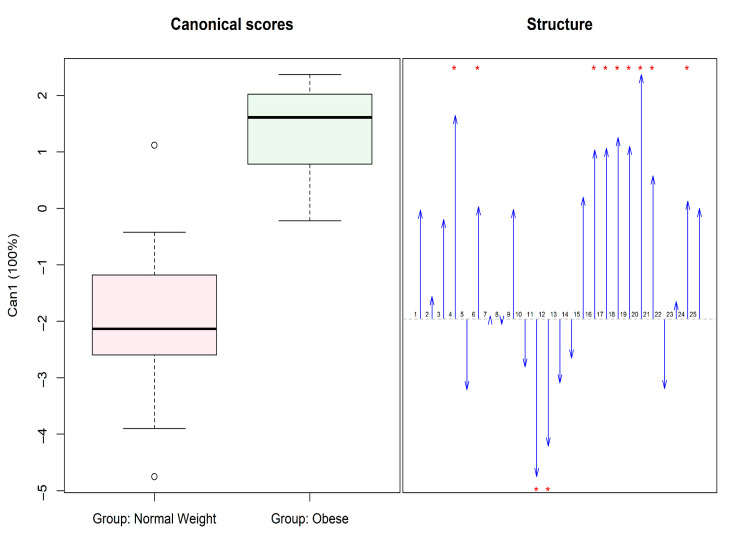
Boxplot of Canonical Scores for FA items and structure for canonical scores. * There are significant differences between the two groups.

**Figure 2 nutrients-17-02654-f002:**
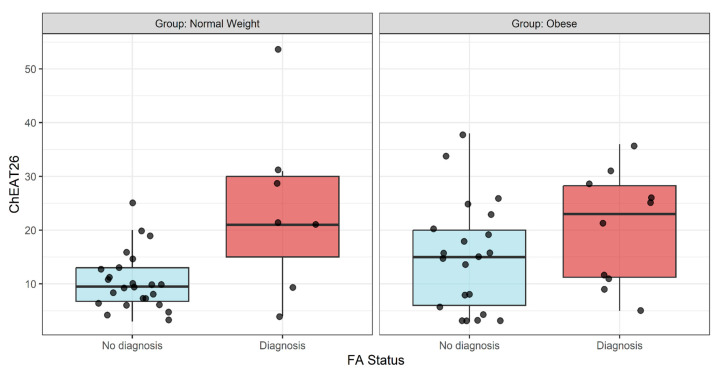
Boxplot of ChEAT-26 scores based on diagnosis, S-YFAS-C diagnosis outcome, and group.

**Table 1 nutrients-17-02654-t001:** Demographic and anthropometric characteristics of the sample by group.

	Group		
	Controls (Normal Weight) n = 31	Cases (Obesity) n = 31	*p*-Value
Male gender, n (%)	16 (51.6)	22 (71.0)	0.118
Age, mean (SD)	10.7 (0.8)	11.0 (0.7)	0.088
Weight (kg), mean (SD)	35.2 (5.4)	73.1 (12.3)	--
Height (m), mean (SD)	1.43 (0.1)	1.55 (0.1)	--
BMI, mean (SD)	17.0 (1.3)	30.1 (3.2)	--
Z-score BMI, mean (SD)	−0.05 (0.52)	2.89 (1.33)	--

BMI: Body Mass Index; SD: standard deviation; kg: kilograms; m: meters.

**Table 2 nutrients-17-02654-t002:** The sample’s score in each outcome measure divided by group.

	Group		Statistic	*p*
	Normal Weight n = 31	Obesity n = 31		
**S-YFAS-C (Food addiction)**				
Quantification of symptoms, mean (SD)	1.61 (1.6)	2.06 (1.7)	−1.07	0.261
Food addiction,* n (%)			0.60	0.437
Yes	7 (22.6)	10 (32.2)		
No	24 (77.4) §	21 (67.7)		
Diagnostic criteria, n (%)				
1. The substance is ingested in a large amount and for longer than expected	2 (6.5)	4 (12.9)	0.74	0.671
2. Chronic desire or repeated attempts to stop consuming it	6 (19.4)	17 (54.8)	8.36	**0** **.004**
3. Time to obtain and use the substance or recovery of its effects	4 (12.9)	5 (16.1)	0.13	>0.99
4. Social, occupational recreational aspects that are abandoned or reduced due to the substance	17 (54.8)	12 (38.7)	1.62	0.203
5. Substance use despite knowledge of adverse consequences	1 (3.6)	8 (25.8)	5.63	**0** **.027**
6. Tolerance	6 (20.7)	8 (25.8)	0.22	0.640
7. Characteristic abstinence symptoms. Substance consumption to relieve abstinence	14 (45.2)	10 (32.3)	1.09	0.297
**ChEAT-26 (Risk of developing an ED)**				
Score, mean (DT)	13.6 (10.5)	16.8 (10.5)	−1.23	0.222
ED risk,** n (%)	7 (22.6)	12 (38.7)	1.90	0.168
**CDRS (Contour drawing rating scale)**				
Perceived body image, median (P25; P75)	4 (3.5, 5)	7 (7, 8)	35.5	**<** **0** **.001**
Desired body image, median (P25; P75)	4 (3, 5)	5 (5, 5)	273	**<** **0** **.001**
Discrepancy in body image, mean (SD)	−0.6 (1.2)	−2.5 (1.2)	102	**<0.001**
Body image dissatisfaction,*** n (%)			34.30	**<0.001**
Yes	3 (9.7)	26 (83.9)		
No	28 (90.3)	5 (16.1)		

§ To avoid sample loss, the response of a participant from the control group was imputed for item 15 of the Y-FAS-C questionnaire based on the most probable response criterion. S-YFAS-C’s 5th and 6th criteria were calculated with a sample of 59 and 60 subjects, respectively. * Food addiction: the existence of 3 or more symptoms on the S-YFAS-C scale and anxiety or significant clinical worsening are present. ** Existence of eating disorder: score ≥ 20 on the ChEAT scale. *** Dissatisfaction with body image: difference ≥ 2 between perceived and desired image. SD: Standard Deviation. When statistical significance exists, the *p* is shaded in bold.

**Table 3 nutrients-17-02654-t003:** Body image dissatisfaction and ED score based on FA diagnosis.

	Normal Weight				Obese				Normal Weight vs. Obese	
	Diagnosis FA n = 7	No Diagnosis FA n = 24	Dif (EE)	*p*-Value	Diagnosis FA n = 10	No Diagnosis FA n = 21	Dif (EE)	*p*-Value	Within FA Diagnosis *p*-Value	Within Non-FA Diagnosis *p*-Value
Discrepancy in body image, mean (SD)	−1.14 (1.86)	−0.48 (0.83)	0.69 (0.48)	0.189	−3.2 (1.23)	−2.19 (1.03)	1.01 (0.42)	**0** **.023**	**0** **.015**	**<0.001**
ChEAT-26 score (ED), mean (SD)	24.14 (16.42)	10.46 (5.39)	13.7 (3.81)	**0** **.001**	20.5 (10.59)	15.1 (10.26)	5.4 (3.98)	0.185	0.585	0.073
S-YFAS-C Symptom count, mean (SD)	4 (1.41)	0.87 (0.83)	3.13 (0.42)	**0** **.001**	4.1 (1.20)	1.1 (0.83)	3 (0.37)	**0** **.001**	0.877	0.239
Z-score IMC, mean (SD)	−0.01 (0.54)	−0.15 (0.48)	0.14 (0.23)	0.552	3.22 (1.74)	2.74 (1.09)	0.48 (0.51)	0.355		

SD: standard deviation; ED: eating disorder; FA: food addiction; ChEAT-26: Children Eating Attitudes Test. Food Addiction diagnosis: the existence of three or more symptoms on the S-YFAS-C scale and anxiety or significant clinical worsening are present. When statistical significance exists, the *p* is shaded in bold.

**Table 4 nutrients-17-02654-t004:** Spearman’s Correlation between scales separated by group.

	Spearman’s R (*p*-Value)		
	S-YFAS-C Quantification of Symptoms	ChEAT-26 Score	Body Image Distortion (CDRS)
ChEAT-26 score			
Overall	0.353 (***p* = 0.005**)	-	
Controls	0.259 (*p* = 0.160)	-	
Cases	0.311 (*p* = 0.088)	-	
Body image discrepancy (CDRS)			
Overall	−0.355 (***p* = 0.005**)	−0.274 (***p* = 0.031**)	-
Controls	−0.264 (*p* = 0.152)	−0.041 (*p* = 0.825)	-
Cases	−0.397 (***p* = 0.027**)	−0.32 (*p* = 0.08)	-

ChEAT-26: Children Eating Attitudes Test; S-YFAS-C: Spanish Yale Food Addiction Scale for Children. When statistical significance exists, the *p* is shaded in bold.

**Table 5 nutrients-17-02654-t005:** Pearson’s correlation between BMI’s z score and outcome measure scores in the overall sample.

	Z Score BMI	
	Spearman’s R	*p*-Value
S-YFAS-C number of symptoms	0.127	0.327
ChEAT-26 score	0.157	0.222
Body image dissatisfaction	−0.677	**<0.001**

ChEAT-26: Children Eating Attitudes Test; S-YFAS-C: Spanish Yale Food Addiction Scale for Children; BMI: Body Mass Index; Contour drawing rating scale. When statistical significance exists, the *p* is shaded in bold.

## Data Availability

All data are available in the manuscript. No artificial intelligence was used in the development of this manuscript.

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
