# Peer review of "Association Between Childhood Obesity and the Risk of Food Addiction: A Matched Case-Control Study"

_nutrients, 2025, doi:10.3390/nu17162654_

Round 1
Reviewer 1 Report
Comments and Suggestions for Authors
This study investigated the association between obesity and food addiction (FA) in a Spanish pediatric population and examined the links between FA and eating disorders (ED) as well as body image dissatisfaction. However, several limitations regarding sample representativeness, methodological details, interpretation of findings, and discussion of limitations need to be addressed:
1. The questionnaire design should be detailed to enhance methodological transparency. In addition, the definition and scope of “junk food” must be clearly specified.
2. Inconsistent data collection: anthropometric data for the case group were collected by medical staff, whereas those for the control group were obtained by school nutritionists, potentially introducing measurement bias.
3. Contradictory results are not adequately explained: e.g., MANOVA indicated an overall significant difference on the S-YFAS-C scale, yet no group differences were observed for individual symptom counts or FA diagnostic rates. How can this discrepancy be interpreted?
4. The manuscript emphasizes the significant correlations between FA symptom counts and ED risk or body dissatisfaction (Table 4); however, the correlation coefficients are low and potential confounders (e.g., psychosocial factors) were not controlled for.
5. The sample was recruited only from Tenerife, Spain, and the age range was limited (9–12 years), which may restrict the generalizability of the findings to other regions or age groups.
Author Response
Reviewer 1:
This study investigated the association between obesity and food addiction (FA) in a Spanish pediatric population and examined the links between FA and eating disorders (ED) as well as body image dissatisfaction. However, several limitations regarding sample representativeness, methodological details, interpretation of findings, and discussion of limitations need to be addressed:
Response:
Dear reviewer, thank you very much for your annotations. We will proceed to answer you on each point.
- The questionnaire design should be detailed to enhance methodological transparency. In addition, the definition and scope of “junk food” must be clearly specified.
Response:
We have expanded the description of the questionnaire administration procedure to enhance methodological clarity. Specifically, we now clarify that the 25 items of the S-YFAS-C were administered exactly as presented in the validated Spanish version, with no modifications to their content or structure. Additionally, we explain that the standard instructions were read aloud to all participants by the researchers, ensuring uniformity in delivery.
Moreover, we have now explicitly defined the term “junk food” as used in our verbal instructions to the participants. This includes examples such as sweets (candy, ice cream, chocolate, cake), carbohydrates (white bread, pasta, rice), salty snacks (chips, crackers), fatty foods (hamburgers, pizza, bacon, steak), and sugary beverages (soda, juice, milkshakes). This clarification has been added to the Methods section (Subsection 2.3: Outcome Measures).
- Inconsistent data collection: anthropometric data for the case group were collected by medical staff, whereas those for the control group were obtained by school nutritionists, potentially introducing measurement bias.
Response:
We appreciate the reviewer’s concern regarding potential measurement bias due to the difference in personnel involved in the anthropometric assessments. Although it is true that medical staff performed the measurements in the case group and school-based nutritionists in the control group, we would like to clarify that the same bioimpedance scale model (Tanita BC-418) was used in both settings, following standardized measurement protocols. This ensured consistency in the data collected. Given that both professional teams were trained in anthropometric assessment and used the same equipment, we consider that the reliability of the measurements was adequately preserved, and the risk of systematic bias was minimized.
- Contradictory results are not adequately explained: e.g., MANOVA indicated an overall significant difference on the S-YFAS-C scale, yet no group differences were observed for individual symptom counts or FA diagnostic rates. How can this discrepancy be interpreted?
Response:
Thank you for highlighting this important point. We have now addressed this issue in the Discussion section of the revised manuscript. Although no statistically significant differences were found in the total symptom count or diagnostic rates, the MANOVA revealed significant differences in the overall response patterns to the S-YFAS-C items. This indicates that children with obesity may exhibit a distinct behavioural profile related to food consumption, even if those differences are not fully reflected in the summary scores.
Importantly, the scoring algorithm of the S-YFAS-C is known to be relatively restrictive, especially in pediatric populations, which may contribute to floor effects and limit the sensitivity of the final diagnosis or symptom count. To account for this potential limitation, we used a MANOVA to explore whether emerging behavioural differences could be detected across the full set of responses. The significant results obtained suggest that certain patterns may indeed be developing in the obesity group, though they may not yet reach the conventional diagnostic threshold. We believe this multivariate approach adds value by capturing early and nuanced divergences in FA-related behaviour.
- The manuscript emphasizes the significant correlations between FA symptom counts and ED risk or body dissatisfaction (Table 4); however, the correlation coefficients are low and potential confounders (e.g., psychosocial factors) were not controlled for.
Response:
Thank you for this insightful observation. We agree that the correlation coefficients reported in Table 4 are statistically significant but of modest magnitude. We have now clarified this in the Discussion section to avoid overinterpretation of the results.
Additionally, we acknowledge that the study did not include control variables such as psychosocial or emotional factors (e.g., impulsivity, self-esteem, family dynamics), which may influence the associations observed. This was due in part to the exploratory nature of the study and the limited sample size. Additionally, we clarified that some of the findings may be artifactual. We have included this point as a limitation.
- The sample was recruited only from Tenerife, Spain, and the age range was limited (9–12 years), which may restrict the generalizability of the findings to other regions or age groups.
Response:
Thank you for this important remark. We fully agree that the geographic and age constraints of the sample may limit the external validity of our findings. We have now included this as an additional limitation in the Discussion section, noting that the results should be interpreted with caution when considering other regions, cultural settings, or age groups. We have also highlighted the need for future studies to replicate these findings in broader and more diverse populations.
Reviewer 2 Report
Comments and Suggestions for Authors
This manuscript addresses an important and timely topic: the relationship between childhood obesity and food addiction, including associations with eating disorders and body image dissatisfaction. The case-control design and the use of validated tools such as the S-YFAS-C and ChEAT are commendable. The study is clearly structured and ethically conducted. However, several aspects of the manuscript would benefit from improved clarity, precision in writing, and more critical interpretation of the findings. Suggestions are provided below by section to improve the overall quality and impact of the manuscript.
introduction
The Introduction provides a relevant and timely overview of food addiction (FA), particularly in the pediatric population. The authors address the conceptual ambiguity of FA and link it well with eating disorders and obesity. The rationale for the study is clear, and the research objectives are outlined. However, the section would benefit from improved linguistic clarity, avoidance of repetition (e.g., “new concept”), and a more structured presentation of the study’s hypotheses. Strengthening the flow and refining some terminology will enhance its academic quality and readability.
Lines 40-42. The phrase “this concept is not yet annexed to any version of the DSM…” is awkward and unidiomatic. Consider rephrasing to: “This concept is not yet officially recognized in the Diagnostic and Statistical Manual of Mental Disorders (DSM) or in the International Classification of Diseases (ICD-11).”
Lines 46-48. The sentence “FA also shares similarities with other substance use disorders, presenting a challenge…” could be made more precise. Consider: “FA displays behavioral patterns similar to substance use disorders, challenging traditional views on eating behavior and dietary self-regulation.”
Line 51. The wording “a questionnaire of questions” is redundant. Recommend revising to: “…a questionnaire that operationalizes the concept of FA by adapting the diagnostic criteria of substance use disorders.”
Lines 58-60. The sentence “This connection underlines the complex interplay…” is good but could benefit from more specificity. Suggest adding a brief example of one such biological or psychological factor, if available from cited literature.
Lines 63-64. Consider rephrasing: “…how addictive behaviors around food consumption may exacerbate or contribute to the development of eating disorders and obesity.”
Methods
The methodology is clearly described, with appropriate inclusion/exclusion criteria, validated tools for assessment, and ethical compliance. The case-control design is well justified, although the imbalance between groups (71 cases vs. 31 controls) and the matching procedure could limit generalizability. The use of standardised instruments such as the S-YFAS-C, ChEAT, and CDRS enhances the reliability of the measures. Statistical methods are generally appropriate, although more detail could be provided regarding how assumptions (normality, etc.) were tested and how missing data, if any, were handled. The ethical procedures are well addressed and comply with international standards.
Lines 142-143. A one-way MANOVA is appropriate for analyzing multiple dependent variables. Please confirm whether assumptions of MANOVA (e.g., multivariate normality, homogeneity of variance-covariance matrices) were tested.
Line 157. Was there any non-response or dropout rate related to consent collection, particularly in the control group? If so, it would be useful to discuss how this may have affected representativeness.
Discussion
The discussion provides a coherent synthesis of the study’s findings and relates them appropriately to the existing literature. The authors effectively highlight the potential implications of food addiction in relation to childhood obesity and eating disorders. Importantly, the limitations of the study are clearly acknowledged, which adds transparency and scientific rigor to the manuscript. However, the discussion would benefit from slightly deeper integration of neurobiological mechanisms and more precise referencing when asserting the novelty of the findings. Overall, the section is well written and contributes meaningfully to the interpretation of the results.
Lines 270-271. Consider adding a brief mention of neurobiological mechanisms, such as dopamine dysregulation, to strengthen the explanation of how ultra-processed foods may promote addictive behaviors.
Author Response
Reviewer 2:
This manuscript addresses an important and timely topic: the relationship between childhood obesity and food addiction, including associations with eating disorders and body image dissatisfaction. The case-control design and the use of validated tools such as the S-YFAS-C and ChEAT are commendable. The study is clearly structured and ethically conducted. However, several aspects of the manuscript would benefit from improved clarity, precision in writing, and more critical interpretation of the findings. Suggestions are provided below by section to improve the overall quality and impact of the manuscript.
Response:
Dear reviewer, thank you very much for your compliments to our article. We will now proceed to respond to each point you have specified.
Introduction
The Introduction provides a relevant and timely overview of food addiction (FA), particularly in the pediatric population. The authors address the conceptual ambiguity of FA and link it well with eating disorders and obesity. The rationale for the study is clear, and the research objectives are outlined. However, the section would benefit from improved linguistic clarity, avoidance of repetition (e.g., “new concept”), and a more structured presentation of the study’s hypotheses. Strengthening the flow and refining some terminology will enhance its academic quality and readability.
Response:
Again, thank you for the recommendations for improvement. In addition to rewording several of the sections, we have proceeded to use more synonyms to avoid repetition of words, as you detail with ‘concept’.
Lines 40-42. The phrase “this concept is not yet annexed to any version of the DSM…” is awkward and unidiomatic. Consider rephrasing to: “This concept is not yet officially recognized in the Diagnostic and Statistical Manual of Mental Disorders (DSM) or in the International Classification of Diseases (ICD-11).”
Response:
Thank you. We have proceeded to make the change you have asked us to make.
Lines 46-48. The sentence “FA also shares similarities with other substance use disorders, presenting a challenge…” could be made more precise. Consider: “FA displays behavioral patterns similar to substance use disorders, challenging traditional views on eating behavior and dietary self-regulation.”
Response:
We have proceeded to make the change you have asked us to make.
Line 51. The wording “a questionnaire of questions” is redundant. Recommend revising to: “…a questionnaire that operationalizes the concept of FA by adapting the diagnostic criteria of substance use disorders.”
Response:
We have proceeded to make the change you have asked us to make.
Lines 58-60. The sentence “This connection underlines the complex interplay…” is good but could benefit from more specificity. Suggest adding a brief example of one such biological or psychological factor, if available from cited literature.
Response:
Thank you again. In this case, we believe that in the previous paragraph, we established which pathologies are most related to these factors.
Lines 63-64. Consider rephrasing: “…how addictive behaviors around food consumption may exacerbate or contribute to the development of eating disorders and obesity.”
Response:
We have proceeded to make the change you have asked us to make.
Methods
The methodology is clearly described, with appropriate inclusion/exclusion criteria, validated tools for assessment, and ethical compliance. The case-control design is well justified, although the imbalance between groups (71 cases vs. 31 controls) and the matching procedure could limit generalizability. The use of standardised instruments such as the S-YFAS-C, ChEAT, and CDRS enhances the reliability of the measures. Statistical methods are generally appropriate, although more detail could be provided regarding how assumptions (normality, etc.) were tested and how missing data, if any, were handled. The ethical procedures are well addressed and comply with international standards.
Lines 142-143. A one-way MANOVA is appropriate for analyzing multiple dependent variables. Please confirm whether assumptions of MANOVA (e.g., multivariate normality, homogeneity of variance-covariance matrices) were tested.
Response:
We thank you for your insightful comment regarding the MANOVA assumptions. Your point prompted us to conduct a more thorough re-evaluation, and we have revised the manuscript substantially to address these crucial issues with full transparency. In our re-assessment, we confirmed two key points:
- Homogeneity of Covariances: We determined that Box's M test cannot be computed for our dataset. This is a known limitation that occurs when the number of observations in a group is smaller than the number of dependent variables. To address this, we have now performed Levene's tests on each dependent variable as a supplementary check and reported these results.
- Multivariate Normality: As you noted, the normality assumption was violated (p < .001), with a visual inspection suggesting a light-tailed distribution.
Given these limitations, we have taken two important steps, which are now detailed in the manuscript. First, we have based our primary inference on Pillai's Trace, which the literature consistently identifies as the most robust statistic to violations of both homogeneity of covariances and normality. Second, we have added a comprehensive discussion in the "Limitations" section to transparently acknowledge these constraints and call for caution in the interpretation of the results.
We believe this revised approach of using supplementary univariate tests and relying on the most robust test statistic is the most
Line 157. Was there any non-response or dropout rate related to consent collection, particularly in the control group? If so, it would be useful to discuss how this may have affected representativeness.
Response:
Thank you for your observation. We confirm that all children who were invited to participate in the study returned signed informed consent forms and agreed to take part. Therefore, no non-response or dropout occurred in either group during the recruitment phase. This information has now been clarified in the revised manuscript (Section 2.1). Additionally, we have provided details regarding the initial sample size and the age-based 1:1 matching process used to create the final analytic sample of 31 cases and 31 controls (Section 2.2).
Discussion
The discussion provides a coherent synthesis of the study’s findings and relates them appropriately to the existing literature. The authors effectively highlight the potential implications of food addiction in relation to childhood obesity and eating disorders. Importantly, the limitations of the study are clearly acknowledged, which adds transparency and scientific rigor to the manuscript. However, the discussion would benefit from slightly deeper integration of neurobiological mechanisms and more precise referencing when asserting the novelty of the findings. Overall, the section is well written and contributes meaningfully to the interpretation of the results.
Lines 270-271. Consider adding a brief mention of neurobiological mechanisms, such as dopamine dysregulation, to strengthen the explanation of how ultra-processed foods may promote addictive behaviors.
Response:
The comment is appreciated and could contribute significantly to the enrichment of the debate. However, given that the aspects mentioned were not dealt with in depth in the framework of the research, and considering that the available data do not allow inferences to be drawn from the analyses carried out, it is not considered appropriate to incorporate this information.
Reviewer 3 Report
Comments and Suggestions for Authors
The paper fits the scope of the journal as it deals with a topic, which is fully in line with the field of nutrition and related health status of individuals.
The methodology of the work is quite simple, yet straightforward, and probably could have gone more in depth when it comes to the clinical scales employed for the characterization of the conditions studied; however, this point could be briefly mentioned in the study limitations.
The main point concerning the article quality is that a more in-depth insight on psychophysiological basis for FA in obese individuals could be useful to better outline the framework of the subject investigated.
As for the methodology, please mention how the calculation of the sample size has been performed.
Furthermore, future investigations derived from the current study can be acknowledged.
Finally, typos should be double checked throughout the manuscript.
Author Response
Reviewer 3:
The paper fits the scope of the journal as it deals with a topic, which is fully in line with the field of nutrition and related health status of individuals.
The methodology of the work is quite simple, yet straightforward, and probably could have gone more in depth when it comes to the clinical scales employed for the characterization of the conditions studied; however, this point could be briefly mentioned in the study limitations.
The main point concerning the article quality is that a more in-depth insight on psychophysiological basis for FA in obese individuals could be useful to better outline the framework of the subject investigated.
As for the methodology, please mention how the calculation of the sample size has been performed.
Response:
Thank you very much for the review and for the comments raised.. We acknowledge that no formal sample size calculation was conducted for this exploratory case-control study. The number of participants was determined pragmatically: cases were recruited consecutively as they attended the paediatric nutrition clinic during the recruitment period, which coincided with the lead researcher’s placement at the hospital. For the control group, we included as many eligible children as possible from a private school that agreed to collaborate with the study. We have now clarified this point by adding a note in the discussion section, specifically within the limitations paragraph.
Furthermore, future investigations derived from the current study can be acknowledged.
Response:
Thank you again. We have added, along with improved wording of the limitations, the need for further research: “Therefore, the conclusions drawn are initial, and a study with the same approach, but with a more representative sample, should be conducted in order to confirm these preliminary results.”
Finally, typos should be double checked throughout the manuscript.
Response:
Thank you. We have proceeded again to revise the errata.
Round 2
Reviewer 1 Report
Comments and Suggestions for Authors
The authors have addressed my comments and revised their manuscript carefully, and the manuscript can be accepted in its present form.
Author Response
Thank you very much for your reviews and your words
Reviewer 2 Report
Comments and Suggestions for Authors
Dear Authors,
Please see below my comments for this manuscript.
Thank you for the opportunity to review the revised version of this manuscript. The authors have addressed the majority of my comments thoroughly and thoughtfully. I particularly appreciate the improvements in linguistic clarity, the transparent handling of statistical assumptions related to the MANOVA, and the more structured presentation of the study’s hypotheses.
One point that remains partially unaddressed relates to the discussion of potential neurobiological mechanisms, such as dopaminergic dysregulation, that may underlie addictive eating behaviors. While I understand the authors’ rationale, I still believe that a brief mention supported by existing literature (e.g., on the reward response to ultra-processed foods) could further enhance the discussion and provide a more comprehensive interpretation of the findings.
Overall, the manuscript has improved considerably and makes a valuable contribution to the literature.
Regards,
Author Response
Our research team sincerely appreciates your valuable feedback and we are very grateful that you have appreciated the changes we have made to the manuscript.
We also agree on the importance of considering the possible neurobiological mechanisms involved, such as dopaminergic dysregulation.
In this sense, we recognize that it would be possible to expand the discussion around the reward response associated with the consumption of ultra-processed foods. However, the space constraints of the manuscript, the limited time allowed for this new review (72 hours), and the specific focus of the study prevent us from developing this aspect in greater depth. Likewise, we consider that this is a field still evolving, characterized by important methodological challenges and conceptual ambiguities that make it difficult to treat it rigorously within the limits of this study.
In fact, we consider that throughout the introduction and in the discussion, we clearly detailed that addictive eating behaviors are influenced by neurobiological mechanisms similar to those of substance addictions. However, not only dysregulation of the dopaminergic system, especially in the mesolimbic reward circuitry, is one of the hypotheses. The endocannabinoid system, which enhances the pleasure associated with food and further stimulates dopamine release, could also be involved. In addition, other neurotransmitters such as serotonin, noradrenaline, GABA and glutamate also play an important role, affecting appetite control, impulsivity and stress response. There are even alterations in hormones and neuropeptides such as leptin, ghrelin and NPY, which regulate hunger and satiety. All this contributes to a feedback loop where immediate pleasure overcomes rational control, perpetuating addictive behavior even in the face of negative consequences.
However, due to the limited dimensions of the manuscript and the difficulties of providing a relevant and well-supported approach to all these hypotheses -including their respective bibliographical references-, it is complex to address them without the content becoming excessively broad in relation to the main focus of the paper.
For this reason, in addition to the MANOVA analysis approach, we substantially modified the paragraph, starting at line 360. This modification was not only to improve the understanding of the MANOVA analysis, as we have detailed, but also to relate criterion 2 (persistent desire or repeated attempts to reduce intake) and criterion 5 (continued consumption despite adverse consequences) to the brain's reward circuitry.
Therefore, we have considered leaving the paragraph as follows: “Furthermore, when analyzing the seven diagnostic criteria individually, the most notable differences were found in criterion 2 (persistent desire or repeated attempts to reduce intake) and criterion 5 (continued consumption despite adverse consequences). These behaviours may reflect the habitual intake of ultra-processed foods, designed with industrial formulations that intensely stimulate the brain's reward circuitry, which could contribute to the development of an addictive response, while promoting excess weight due to their high caloric content.22 In fact, these criteria are the ones that usually give the most positive results in studies, as suggested in 2016 by Imperatori C. et al.23”
Again, we are deeply grateful for your understanding. If you feel that this aspect requires further elaboration, we would be willing to expand on it if you have more time available, as we value your observation and share your interest in strengthening the content of the manuscript as much as possible.
